# Eco-Geographical and Botanical Patterns of Resistance to Lepidoptera Insects in *Brassica rapa* L.

**DOI:** 10.3390/plants13050673

**Published:** 2024-02-28

**Authors:** Anna M. Artemyeva, Anastasia B. Kurina

**Affiliations:** Federal Research Center N.I. Vavilov All-Russian Institute of Plant Genetic Resources, 190000 St. Petersburg, Russia; akme11@yandex.ru

**Keywords:** *Brassica rapa* L. crops, biodiversity, diamondback moth *Plutella xylostella*, cabbage moth *Mamestra brassicae*, herbivore pest resistance, geographical trials

## Abstract

In the context of the widespread expansion of damage by herbivorous pests of *Brassica* crops, taking into account the requirements for minimizing pesticide pollution of the environment, it is important to have fundamental knowledge of the geographical features of the distribution of pests and about the botanical confinement of plant resistance in order to develop a strategy for creating new *Brassica* cultivars with complex resistance to insects. The relevance of our work is related to the study of the variability in the degree of resistance of the extensive genetic diversity of *Brassica rapa* accessions to the main herbivorous pests of *Brassica* crops in contrasting ecological and geographical zones of the Russian Federation (Arctic, northwestern, and southern zones). We have studied the distribution and food preferences of Lepidoptera insects (diamondback moth *Plutella xylostella* and cabbage moth *Mamestra brassicae*) on a set of 100 accessions from the VIR *B. rapa* collection (Chinese cabbage, pakchoi, wutacai, zicaitai, mizuna, and leaf and root turnips) in the field in three zones of the Russian Federation. We have found that the diamondback moth and cabbage moth are largely harmful in three zones of the European part of the Russian Federation, although the degree of damage to plants by these insects varies by year of cultivation. On average, for the set studied during the two years of the experiment, the degree of plant damage by both pests in the Arctic zone was low and almost low, and in the northwestern and southern zones, it was medium. It was noted that diamondback moth damage was greater in the northwestern zone in both years and in the southern and Arctic zones in 2021, while in 2022, the degree of cabbage moth damage was slightly higher in the southern and Arctic zones. Under the conditions of field diamondback moth damage, the accessions of Chinese cabbage, wutacai, and mizuna turned out to be the most resistant (the damage score was 1.92–1.99), whereas the accessions of wutacai and pakchoi were the most resistant to the cabbage moth (the damage score was 1.62–1.78). A high variability in the degree of resistance of *Brassica* crops to Lepidoptera insects from complete resistance to susceptibility was revealed. We have identified sources of resistance to insects, including complex resistance in all study areas, among landraces and some modern cultivars of Chinese cabbage, pakchoi, wutacai, and mizuna from Japan and China, as well as European turnips. The highest susceptibility to pests in the studied set was noted in the accession of root turnip “Hinona” (k-1422, USA) (average damage score of 3.24–3.53 points). We were not able to establish the morphological features of resistant plants or the geographical confinement of the origin of resistance of *B. rapa* crop accessions.

## 1. Introduction

*Brassica* vegetable crops are important in human economic activity [1,2]. About 10% of the world’s food is provided by *Brassicaceae* crops, and *B. rapa* crops occupy one of the most important places among them. *B. rapa* crops (Chinese cabbage, pakchoi, wutacai, zicaitai, mizuna, and leaf and root turnips) are early-ripening productive crops characterized by the presence of valuable biochemical compounds and are relatively easy to grow. Chinese cabbage and root turnips are widespread across the globe; other crops are cultivated locally. *B. rapa* crops are represented by large varietal diversity with a wide range of morphological and biological characteristics [3,4].

*Brassica* crops are affected by many types of insect pests, primarily Lepidoptera insects: cabbage moths, diamondback moths, cabbage and turnip butterflies, as well as flies, thrips, and aphids. The study of resistance to these pests is given considerable attention around the world [5]. Lepidoptera pests can cause significant damage to these crops. Damage consists of feeding on the leaves, affecting the value of the crop for human consumption. Damage from Lepidoptera pests could be reduced using resistant cultivars.

The cabbage moth (CM), *Mamestra brassicae* L. (Lepidoptera, Noctuidae), is a generalist insect pest; it has been found feeding on a wide range of vegetable and legume plant species. In total, it has been found on the plants of 22 families [6], but it is believed that it prefers *Brassica* species.

The diamondback moth (DBM), *Plutella xylostella* L. (Lepidoptera, Plutellidae), is a specialized pest of *Brassicaceae* crops that is distributed everywhere. This insect is characterized by a short life cycle; it reproduces very well in warm temperatures but finds it difficult to adapt to cold temperatures [7].

DBM has now acquired the status of the most dangerous pest of *Brassicaceae* crops in the world, including Russia [8,9,10]. In *Brassica* crops, morphological, growth, and physiological barriers against DBM damage have been revealed [11,12,13,14].

Climate changes such as global warming are influencing the reproductive potential of pests, resulting in new or increased insect pest incidence [15].

One of the most important and complex trends in crop breeding is the creation of highly productive cultivars and hybrids with group resistance to pests [11]. The resistance of *Brassica* crops to pest damage should be incorporated into the genotype at the earliest stages of the breeding process [16,17,18].

In agriculture, various methods of plant protection are used, and one of the promising trends is the study of the plant’s own protective mechanisms and the development of genetically resistant cultivars.

Under the influence of negative environmental factors and stressful situations at all stages of the life cycle of a plant, its natural immunity decreases. The stress load on plants depends on the climatic and geographical conditions of cultivation [19,20]. It should be noted that some pest, for example, the diamondback moth, are resistant to many pesticides due to overuse, especially in Asia [21,22].

Although the use of pesticides in cultivation is the most farmer-friendly, easy, and effective method, it may cause ecological damage as well as the chemical contamination of products and the environment. The current strategies of Lepidoptera pest management rely largely on the use of synthetic pesticides, but this approach is undesirable because crops are used for human consumption, and pesticides cause negative effects on the natural enemies of the pest.

Plant–insect control methods have been developed by generating *Bt* gene (*Bacillus thuringiensis*) transgenic crops. However, researchers have argued about the safety of genetically modified crops [23,24].

Although the new crop breeding methods involving natural selection and traditional crossing do not provide a fast or perfect control method like pesticides and transgenic plants, they have been considered to be desirable methods for controlling the infestation of Lepidoptera insects because they are less labor intensive, involve lower costs, and improve food safety.

The first step in developing resistant cultivars is to identify sources of resistance. Insect resistance in *Brassica* crops is well documented, but most studies have focused on resistance to three major Lepidoptera pests: *Pieris rapae* L. (Lepidoptera, Pieridae), *P. xylostella* L., and *Trichoplusia ni* (Lepidoptera, Noctuidae) [5,25,26,27,28,29,30,31,32].

Few studies have been conducted to find *Brassica* germplasm resistant to other important Lepidoptera pests in Europe, such as *M. brassicae* and *Pieris brassicae* L., or to study the seasonal occurrence of these pests [31].

The Russian state global Brassicaceae collection preserved at the Vavilov Institute (VIR) is 100 years old: its formation began in 1923 after N.I. Vavilov’s visit to West-European countries, the USA, and Canada (1921–1922). Later on, collecting missions were conducted in Russia and all centers of plant origin and diversity all over the world.

Presently, the VIR Brassicaceae collection consists of 10,997 accessions of 11 genera and 32 species of vegetable, fodder, oilseed, spicy, and ornamental crops from 98 countries. The vegetable Brassicaceae collection includes 7272 accessions, among which there are 605 of turnip *B. rapa* and 1048 of leafy *B. rapa* (Asian *brassicas* and broccoletto). The oilseed Brassicaceae collection consists of 3725 accessions, including 353 of turnip rape *B. rapa oleifera* Metzg. According to Vavilov, a worldwide collection should reflect the natural biodiversity of cultural crops and their wild relatives and the general gene pool of modern breeding achievements. The collection includes accessions of different statuses: wild species, landraces (30%), old and advanced breeding cultivars (58%), inbred and double haploid lines, hybrid populations, and mapping populations (12%).

The purpose of this work was to study the distribution of the diamondback moth and cabbage moth in three climatically different eco-geographical zones of the Russian Federation to survey preferences of herbivore pests using a set of 100 *B. rapa* accessions representing 6 vegetable crops from the VIR core collection, to search for sources of resistance to *M. brassicae* and *P. xylostella* among *B. rapa* crops and cultivar types, and to create the lines with complex resistance to insects.

## 2. Results

### 2.1. Phytopathological Observations and Accounting

The biological features of the diamondback moth and cabbage moth differ significantly. During the growing season, predominantly 2 generations of DBM develop in the north of the European part of Russia, up to four generations in the south, and 1–2 generations of CM, the first generation being the most harmful, since later on, the number of pests decreases due to often unfavorable weather conditions and the activity of numerous entomophages: *Trichogramma evanescens* West. (Hymenoptera, Trichogrammatidae), *Apanteles glomeratus* L. (Hymenoptera, Braconidae) and others. In DBM, the image is a medium-sized nocturnal butterfly (wingspan 15–18 mm); in CM, it is large (35–40, up to 50 mm) and nocturnal. Oviposition of DBM begins, on average, noticeably earlier than that of the cabbage moth and occurs in the second or third decade of May, while in CM, it takes place in early June. The nature of the oviposition of pests is different: DBM lays 5–15 yellow eggs on the underside of the leaf, and CM lays 30–100 almost transparent eggs in the shape of a polygon.

The junior DBM caterpillar is white with a dark head. The next instars are green and grayish green with dark brown spots on the head and a length of 11–13 mm. The CM caterpillar is gray-green to dark brown, up to 50 mm long.

*Brassicas* pests cause different types of damage to plants. DBM is a leaf-mining pest. DBM caterpillars of the first instar leave a window type of damage, while caterpillars of older instars leave small perforated damage. In CM, the first instar caterpillars skeletonize the leaves; in subsequent instars, they gnaw through perforated holes and roughly eat away at the leaves, heavily polluting the plants with waste from their vital activity.

In Pushkin (northwestern zone), the main pests of *Brassica* crops in 2021 were DBM, and cabbage butterflies, fleas, and thrips were noted to a small extent. In 2022, the number of pests was, on average, significantly less than in 2021, which is most likely due to unfavorable wintering conditions for pests. The main pest in 2022 was DBM, CM was less common, and cabbage butterflies, fleas, and thrips were noted to a very small extent (Figure 1).

In Murmansk (Polar Experiment Station, Arctic zone), *Brassica* crops were affected in 2021 only by DBM. In 2022, *Brassica* crops were damaged mainly by CM. To a lesser extent, damage by DBM and cabbage fly was noted.

In Maykop (southern zone of Russia), the first pests of *Brassica* crops in 2021 were the crucifer flea beetle (from 25 May) and the cabbage butterfly (the first generation was noted on 5 June, and the second on 5 July). The appearance of the first DBM caterpillars was noted on 15 June, and their maximum number was observed on 20–30 June. The colonization of some accessions is very uneven (20–100% of the plants in an accession). On average, 60–80% of plants in an accession were colonized, with 1–3 caterpillars per plant. CM was relatively harmful on 1–15 July, infesting 50–60% of plants, 0.5–1 caterpillars per plant. The damage assessment was carried out on 27–28 July.

In 2022, along with DBM and CM, the crucifer flea beetle and cabbage fly were also significant pests of *Brassica* crops.

On average, for the studied set, damage to plants by DBM was higher than that by CM in Pushkin in 2021–2022 and Maykop and Murmansk in 2021, while in 2022, the degree of CM damage was slightly higher than the degree of DBM damage in Maykop and Murmansk.

### 2.2. Diamondback Moth (Plutella xylostella)

We have established differences in the degree of DBM damage depending on the year and the ecological and geographical zone of cultivation (Figure 2).

On average, for the studied set and for each crop, the degree of damage in Pushkin in 2022 was significantly lower than in the previous year. In Maykop, the same regularity was noted, but significant differences over the years were especially established for pakchoi, wutacai, and mizuna. In Murmansk, differences in the degree of plant damage by years were insignificant (Figure 2).

In 2021, the degree of DBM damage on average for the studied set and for each crop was almost the same in Pushkin and Maykop (score of 2.62–2.65) and significantly lower in Murmansk (score of 1.54) (Figure 2). In 2022, the degree of infection by DBM on average for the studied set was significantly higher in Maykop (medium) and similar in Pushkin and Murmansk (low). At the same time, in accessions of Chinese cabbage, the degree of damage was higher in Murmansk compared to Pushkin, while in accessions of other crops, the degree of damage was lower in Murmansk compared to that in Pushkin. The amplitude of variability in the degree of DBM damage on average for the studied set included all gradations from resistance to susceptibility.

The analysis of variance revealed that the factors of plant genotype and place of cultivation (15.1 and 12.9%) contribute to the formation of resistance, and the interaction of the factors genotype × place and genotype × place × year (20.2 and 16.4%) is more noticeable. The contribution of other factors remains important (22.1%), which may be associated with other stresses (Table 1).

We have identified botanical differences between crops in terms of DBM resistance. On average, Chinese cabbage, wutacai, and mizuna were the most resistant (damage score of 1.92–1.99). Pakchoi, zicaitai, and leafy and root turnips had a high degree of damage (score of 2.21–2.37). The variability in the degree of resistance to DBM damage within a crop was very high in wutacai, pakchoi in Maykop and Murmansk, mizuna, and leaf and root turnips in Murmansk. We have identified accessions that are relatively resistant to DBM under all growing conditions (Table 2).

The turnip root accession “Hinona” (k-1422, USA) was characterized by a high degree of susceptibility to DBM (average damage score of 3.24).

### 2.3. Cabbage Moth (Mamestra brassicae)

We have identified differences in the degree of CM damage depending on the year of cultivation. On average, for the studied set and for each crop, the degree of damage in Pushkin in 2022 was significantly lower than in 2021 by an average of 85% and by 75–125% for individual crops. On the contrary, in Maykop, more serious damage was noted in 2022 than in 2021 on average for the studied set (by 67%), and especially great differences were noted for Chinese cabbage (by 115%) and for pakchoi and wutacai (by 173, 146%, respectively). No CM damage was detected in Murmansk in 2021, while in 2022, the average damage score for the studied set was 1.69, and damage above the average was observed in zicaitai and leaf turnips.

Differences in the degree of CM damage depending on the plant growing zone are not clearly expressed, except for a relatively low degree of damage in Murmansk (Figure 3). Also, the degree of CM damage to mizuna, leaf, and root turnips during the years of cultivation in Maykop was above average.

The analysis of variance revealed that the plant genotype (16.4%) and, to a greater extent, the genotype × place interaction (25.9%) contribute to the formation of resistance. The largest contribution is made by other factors (56.3%), which are associated with other stresses, such as weakening plants and earlier damage by other pests, primarily the diamondback moth (Table 3).

On average, the most resistant accessions were those of pakchoi, wutacai, and Chinese cabbage (1.62–1.81 points), and the least resistant accessions were those of zicaitai, mizuna, leaf and root turnips (2.12–2.43 points).

On average, accessions with an average damage score of less than 1.5 showed high resistance to CM (Table 4).

The highest susceptibility to CM damage was found in accessions of leaf and root turnips from Japan. The root turnip “Hinona” (k-1422) had the highest degree of damage (3.53 points).

### 2.4. Sources of Complex Resistance and Creation of Source Material for Herbivorous Pests

We isolated *B. rapa* accessions with relatively high complex resistance to herbivorous pests during two-year studies in three eco-geographical zones of the Russian Federation. All selected accessions were also characterized by important economically valuable traits: high yield, crop quality, including a high content of biologically active compounds, a relatively compact leaf rosette, and resistance to early stemming (Figure 4).

In 2021–2022, several lines were created by self-pollination of individual resistant plants from highly resistant accessions of Chinese cabbage, pakchoi, wutacai, zicaitai, and leaf turnips with economically important traits. In 2022, F1 hybrids were obtained as a result of hybridization of individual lines. The lines and F1 hybrids were assessed for resistance to DBM and CM on an artificial infectious background in 2023 (Table 5).

Thus, lines and F1 hybrids were obtained under conditions of a strict artificial infectious background and showed almost complete resistance to both pests. The high content of biologically active vitamins and pigments was characteristic of hybrids of old local Japanese forms and hybrids of rosette cabbage and turnip leaf. These hybrids created in the USA are the most valuable turnip crops from the biochemical point of view.

## 3. Discussion

About 10% of the world’s vegetable production is generated from Brassicaceae, wherein *B. rapa* is a dominating species. The species encompasses a multitude of field-grown crops, including several subspecies of *B. rapa*, like leafy vegetables such as heading Chinese cabbage, non-heading pakchoi and Japanese types, the turnips with tuberous hypocotyl, and broccoletto, which inflorescences are consumed.

The most common pests of Brassicaceae crops around the world, including the Russian Federation, are the thrips *Thrips tabaci* L. (Thysanoptera, Thripidae), cabbage fly *Delia radicum* L. (Diptera, Anthomyiidae), cabbage moth (*M. brassicae*) and diamondback moth (*P. xylostella*). In the central and southern regions of Russia, damage from these pests reaches 50% of harvest loss or damage since chemical control measures do not produce a significant effect. The distribution of pests has increased in recent years due to climate change and the expansion of rapeseed. At the same time, the specificity of human consumption of these crops as food requires the maximum degree of ecological safety of products, and modern ecological thinking in plant protection requires minimizing the anthropogenic impact on the components of agrobiocenosis, including useful entomofauna. Therefore, the role of genetic sources of resistance, preserved in ex situ collections, including landraces with high adaptability, is increasing in breeding programs.

For instance, the *Brassica oleracea* L. [33] and rapeseed [34] collections were tested for resistance to *Brevicoryne brassicae* L. (Hemiptera, Aphididae). It was not possible to identify forms of cabbage crops completely resistant to thrips; however, highly tolerant cabbage accessions to tobacco thrips were found [35,36].

Bagrov et al. [37] studied the resistance of white cabbage and Chinese cabbage to diamondback moth and cabbage moth. In this work, white cabbage lines resistant to diamondback moth and “Hydra” (a hybrid of Chinese cabbage) resistant to herbivorous pests were identified and recommended for use to reduce the pesticide load.

The mechanisms of interaction between the *Brassicas* and the cabbage moth were studied by Cartea et al. [38]. It has been shown that the most resistant lines of kale have glossy leaves without a waxy coating.

One of the largest genebanks hosting *Brassica* collections is that at the N.I. Vavilov Institute of Plant Genetic Resources (VIR), Russian Federation. The fundamental objectives of the VIR scientific activities are collection, conservation, and prebreeding evaluation of the accumulated biodiversity under biotic and abiotic stresses in different eco-geographical zones [39], and therefore good prospects of screening these collections for the sources and donors of resistance to Lepidoptera insects are beyond doubt.

The results of long-term field observations of the damage to about 1000 accessions from *Brassica* crop collections by diseases and pests in the Leningrad Province and the problems of protecting plants from a complex of pests are presented in the works of Belykh et al. [40] and Gasich [41]. For instance, 7 types of thrips were recorded on *B. oleracea* crops in the northwestern region. Thrips affect all cole crops, but the greatest harm is committed to white cabbage, up to five covering leaves of which are damaged in the northwestern region. Sources of resistance are found in the Likurishka cultivar of Turkish-Bulgarian origin. Among *B. rapa* crops, the heading Chinese cabbage is mostly damaged by the thrips.

Many years of field research at VIR showed that kale, Brussels sprouts, and leafy rape *Brassica napus* L. var. *pabularia* are highly resistant to all herbivorous pests, especially the genetic sources from the UK, Germany, Sweden, and Canada.

Zakharova et al. [10] conducted a field assessment of *Brassica* accessions regarding colonization and damage of plants by the diamondback moth to identify resistant forms in the VIR collection in the suburbs of St. Petersburg.

Among the Lepidoptera pests, the most damaging now for all *Brassicas* in Russia is the cabbage moth. The diamondback moth damages the leaves and spoils the appearance of the plant, while the cabbage moth severely damages the leaf surface, causing total harvest loss.

White cabbage accessions from Afghanistan, new hybrids of Russian origin; some red cabbage accessions from China, France, and the Netherlands, as well as Savoy cabbage from Russia, the Netherlands, and Japan were low affected by the cabbage moth and diamondback moth. The most serious damage was noted in cauliflower and broccoli accessions, and the most serious damage was to kohlrabi. The variation in resistance levels to the diamondback moth among *B. oleracea* crops was often higher among accessions of the same crop than between crops.

The highest level of resistance to the cabbage moth among *B. rapa* crops was observed in VIR for some years in Chinese cabbage accessions, their pubescence (trichomes) presumably being among the reasons. A lower but still high level of resistance was observed in mizuna, a medium level in zicaitai, a medium and somewhat lower level in tatsoi and leafy turnip, and susceptibility was observed in pakchoi. The damage of *B. rapa* crops by the diamondback moth was weak, so we could not find any patterns.

At the same time, we previously noted that the number of pests and, accordingly, damage to plants increases significantly with the advancement from the north of Russia to the south. Therefore, it was important to conduct a comparative analysis of pest resistance in contrasting climatic zones, and it was the reason for the current study.

Some tatsoi and turnip accessions showed the biggest resistance towards herbivore feeding. One of the most important breeding areas is the creation of highly productive cultivars and hybrids of *Brassica* with resistance to pests. The resistance of plants to damage by pests should be incorporated in the genotype at the very early stages of the breeding process. Since the stress load on plants depends on the weather and climatic conditions of the geographical zone and the year of cultivation, it is necessary to conduct an annual assessment of plant performance at different locations. We selected resistant accessions based on field experiments, identified inbred lines among them, and made laboratory and field tests. These new lines were included in hybridization breeding programs.

## 4. Materials and Methods

Field resistance to herbivorous pests of a representative set of 100 accessions from the VIR collection, representing the diversity of five subspecies and six crops of *B. rapa*, was assessed in two consecutive years (2021 and 2022). The set included Chinese cabbage (40 accessions), pakchoi (21), wutacai (7), zicaitai (1), mizuna (5), leaf (6), and root turnip (20 accessions). The plants were grown using the same methodological approach at three experiment stations of VIR located in contrasting eco-geographical zones of the Russian Federation: in the Murmansk Province (68° 58′ N, 33° 5′ E), north of the Arctic Circle, in Pushkin (59° 43′ N, 30° 25′ E) near St. Petersburg in the northwestern region, and in Maykop (44° 36′ N, 40° 5′ E), in the southern zone near the Black Sea. The experiment was performed in two replications, 30 plants per replication, against a background that, during a period from 1 to 20 June, was provocative in terms of cabbage moth and diamondback moth damaging activity. Assessment of the damage was carried out 3 times a week, with the final assessment at full technical maturity of plants, but not later than 40 days after sprouting or at the moment of maximum damage of leaf lamina. 

The study of the resistance of lines and hybrids to artificial infestation by pests was carried out in the greenhouse conditions of the Pushkin Laboratories of VIR using the method of Hoang et al. [42] in 2023. The experiment was carried out in three replications, 20 plants per replication.

The following scores were used for assessing pest resistance/susceptibility in plants [43,44]:

Damage    % Leaves score     damaged

0—    no leaf damage or up to 5%

1—     6 to 25%

2—    26 to 50%

3—    51 to 75%

4—    76% to 100%.

The average damage score for each accession was calculated as the average damage of all plants in the accession.

### Statistical Analysis

The obtained data were analyzed using STATISTICA v.12.0 software (StatSoftInc., Tulsa, OK, USA). The data were analyzed by descriptive statistics, the means of normally distributed data were compared using the multivariate analysis of variance (MANOVA), and testing of data for normal distribution was performed using the Shapiro–Wilk test and quantile-quantile plot (QQ Plot). The Tukey HSD (honestly significant difference) post hoc test and LSD_05_ (Fisher’s Least Significant Difference) were used to identify the differences between the means for each characteristic. The single-factor and multivariate analysis of variance (MANOVA) was used to identify the influence of genotype and environmental conditions on traits.

## 5. Conclusions

It has been established that DBM and CM are largely harmful in three zones of the European part of the Russian Federation, although the degree of damage to plants by them varies over the years of research. On average, for the set of accessions studied during the two years of the experiment, the degree of damage to plants by both pests in the northern zone was weak and close to weak, and medium in the northwestern and southern zones. It was noted that DBM damage was greater than that by CM in the northwestern zone in both years, in the southern and northern zones in 2021, while in 2022, the degree of CM damage was slightly higher than the degree of DBM damage in the southern and northern zones.

Under conditions of natural infection with DBM during the years of research, accessions of Chinese cabbage, wutacai, and mizuna turned out to be the most resistant (average score of damage to the crop was 1.92–1.99). Accessions of wutacai and pakchoi are resistant to CM (average score of damage was 1.62–1.78). A high variability in the degree of resistance of *Brassica* crops to Lepidoptera insects, from complete resistance to susceptibility, was revealed. Sources of resistance to insects, including complex resistance, were identified in all study areas, among Chinese and Japanese landraces and individual modern cultivars of Chinese cabbage, pakchoi, wutacai, and turnip. The highest susceptibility to pests in the studied set was observed in the root turnip “Hinona” (k-1422, USA) (average damage score of 3.24–3.53). Lines and F1 hybrids resistant to herbivorous pests have been created.

## Figures and Tables

**Figure 1 plants-13-00673-f001:**
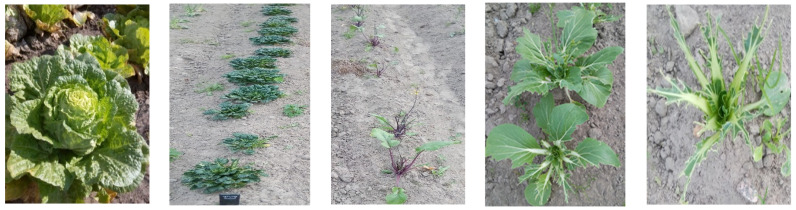
Phenotypic appearance of *B. rapa* subspecies regarding preferences of herbivore pests in Pushkin. From left to right: Chinese cabbage (0–0.5), tatsoi (1.5), zicaitai (2.0), pakchoi (2.5), pakchoi (4.0). Numbers in parentheses represent resistance indices.

**Figure 2 plants-13-00673-f002:**
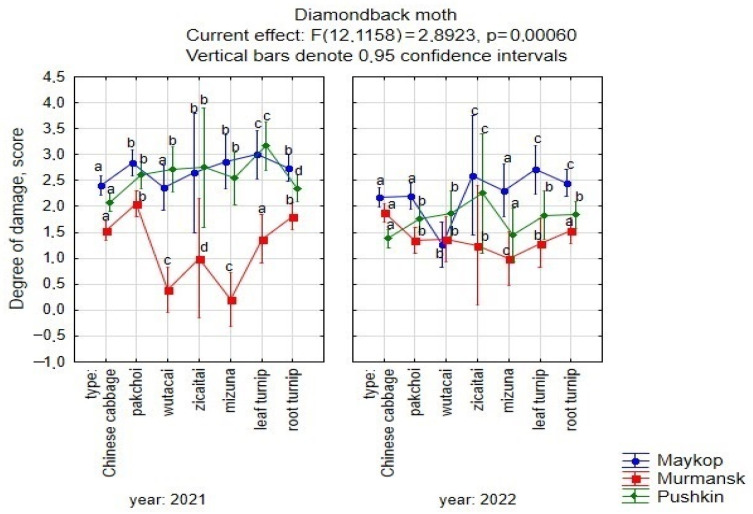
The degree of DBM damage to crops under different growing conditions. ^a–d^ Values with different superscripts were significantly different (*p* < 0.05).

**Figure 3 plants-13-00673-f003:**
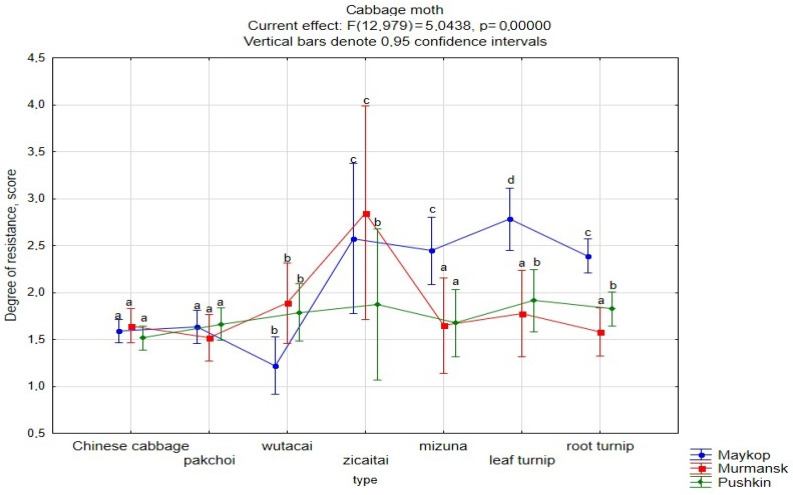
The degree of CM damage to crops under different growing conditions (2022). ^a–d^ Values with different superscripts were significantly different (*p* < 0.05).

**Figure 4 plants-13-00673-f004:**
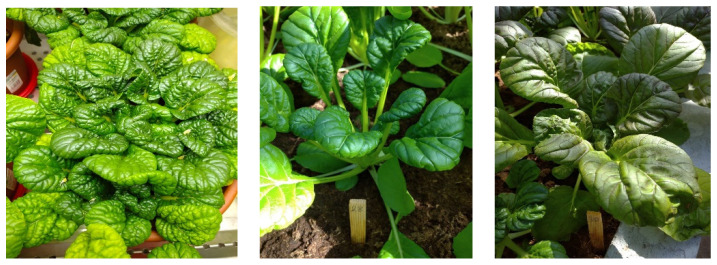
*Brassica rapa* accessions with high complex resistance to herbivorous pests: wutacai “Xiaoba Ye Ta Cai” (k-695); wutacai “Chiimi Jukina” (k-271); F1 “Red tatsoi × Bansei Komatsuna”.

**Table 1 plants-13-00673-t001:** Significance and contribution of factors to DBM resistance (MANOVA, *p* ≤ 0.01).

Factors	SS	Degr. of	MS	F	Contribution, %
Genotype	172.51	99	1.74	4.13	15.1
Place	148.26	2	74.13	175.91	12.9
Year	42.08	1	42.08	99.84	3.7
Genotype × place	232.03	198	1.17	2.78	20.2
Genotype × year	78.04	99	0.79	1.87	6.8
Place × year	32.14	2	16.07	38.14	2.8
Genotype × place × year	188.18	198	0.95	2.26	16.4
Other	252.85	600	0.42		22.1
Total	1146.07				

SS—sum of squares; Degr. of—degrees of freedom; MS—mean squares; F—F-value.

**Table 2 plants-13-00673-t002:** *Brassica rapa* accessions with a relatively high degree of DBM resistance (2021–2022).

№	Catalog Number	Name, Origin	Degree of Damage, Score			
			Pushkin	Maykop	Murmansk	Mean
Chinese cabbage						
1	k-270	Harumaki Shin Santousai, Japan	1.8 ± 0.4 ^b^	2.5 ± 0.3 ^d^	1.4 ± 0.5 ^c^	1.9 ± 0.6 ^c^
2	k-247	Kasinbechu, South Korea	1.5 ± 0.4 ^b^	2.4 ± 0.6 ^d^	1.5 ± 0.3 ^c^	1.8 ± 0.6 ^c^
3	k-292	Ju Zhu, China	1.3 ± 0.6 ^a^	1.7 ± 0.2 ^b^	2.2 ± 0.8 ^d^	1.7 ± 0.7 ^c^
4	k-406	Shantung Tropical Round, Japan	1.6 ± 0.3 ^b^	2.0 ± 0.0 ^c^	1.5 ± 0.5 ^c^	1.7 ± 0.4 ^c^
5	vr.k-1557	Mini Raioh 50 F1, Japan	1.3 ± 0.2 ^a^	1.6 ± 0.2 ^b^	1.4 ± 0.5 ^c^	1.5 ± 0.3 ^b^
6	vr.k-1558	Raioh 90 (F1), Japan	1.3 ± 0.3 ^a^	2.2 ± 0.2 ^c^	1.4 ± 0.6 ^c^	1.6 ± 0.5 ^b^
7	vr.k-1574	Megapolis F1, Russia	1.2 ± 0.6 ^a^	1.6 ± 0.2 ^b^	1.5 ± 0.5 ^c^	1.5 ± 0.5 ^b^
Mean			1.7 ± 0.7	2.3 ± 0.5	1.7 ± 0.7	1.9 ± 1.0
Pakchoi						
8	k-195	Local, China	1.3 ± 0.4 ^a^	2.0 ± 0.0^c^	1.5 ± 0.5 ^c^	1.6 ± 0.4 ^b^
9	k-332	Chingensay, Japan	1.6 ± 0.2 ^b^	1.8 ± 0.9 ^b^	1.6 ± 0.4 ^c^	1.7 ± 0.5 ^c^
Mean			2.2 ± 0.7	2.5 ± 0.6	1.7 ± 0.6	2.1 ± 0.7
Wutacai						
10	k-271	Chiimi Jukina, Japan	2.1 ± 0.7^c^	1.0 ± 0.3 ^a^	0.7 ± 0.2 ^a^	1.3 ± 0.9 ^a^
11	k-649	Tatsoi, Japan	1.8 ± 0.2 ^b^	2.1 ± 0.7 ^c^	1.0 ± 0.4 ^b^	1.6 ± 0.8 ^b^
12	k-695	Xiaoba Ye Ta Cai, China	1.5 ± 0.8 ^b^	1.0 ± 0.3 ^a^	0.5 ± 0.2 ^a^	1.0 ± 0.5 ^a^
13	vr.k-1409	Red Tatsoi F1, USA	2.3 ± 0.4 ^c^	1.1 ± 0.4 ^a^	0.9 ± 0.4 ^b^	1.5 ± 0.8 ^b^
Mean			2.3 ± 0.9	1.8 ± 1.0	1.8 ± 1.0	1.7 ± 1.0
Mizuna						
14	k-434	Shirojuki Sensuji Kyo Mizuna, Japan	1.5 ± 0.6 ^b^	2.3 ± 0.2 ^c^	0.2 ± 0.1 ^a^	1.4 ± 0.8 ^b^
15	k-681	Xiye Xuelihong, China	2.0 ± 0.3 ^c^	2.5 ± 0.1 ^d^	1.0 ± 0.3 ^b^	1.6 ± 0.9 ^b^
Mean			2.0 ± 0.6	2.6 ± 0.5	0.6 ± 0.3	1.7 ± 1.0
Root turnip						
16	k-1050	Local, Russia	1.6 ± 0.2 ^b^	2.4 ± 0.3 ^d^	1.3 ± 0.2 ^c^	1.8 ± 0.5 ^c^
17	k-1398	Palitra, Russia	1.7 ± 0.2 ^b^	2.0 ± 0.0 ^c^	1.3 ± 0.4 ^c^	1.7 ± 0.3 ^c^
18	k-1405	Ova daehnfeldt, Denmark	1.3 ± 0.2 ^a^	2.1 ± 0.2 ^c^	1.4 ± 0.5 ^c^	1.7 ± 0.4 ^c^
19	k-1424	Golden Ball, Finland	1.8 ± 0.2 ^b^	1.9 ± 0.2 ^b^	1.3 ± 0.3 ^c^	1.6 ± 0.3 ^b^
Mean			2.1 ± 0.7	2.6 ± 0.6	1.7 ± 0.7	2.1 ± 0.7
Mean for the studied set			2.0 ± 0.7	2.4 ± 0.6	1.6 ± 1.2	2.0 ± 0.9
LSD_05_			0.19	0.14	0.32	0.27

All data are presented as mean ± SD. ^a–d^ Values with different superscripts in the column differed significantly (*p* < 0.05). LSD_05_—Fisher’s Least Significant Difference.

**Table 3 plants-13-00673-t003:** Significance and contribution of factors to CM resistance (MANOVA, *p* ≤ 0.01).

Factors	SS	Degr. of	MS	F	Contribution, %
Place	10.05	2	5.02	8.59	1.4
Genotype	120.06	98	1.23	2.09	16.4
Place × genotype	189.05	196	0.97	1.65	25.9
Other	411.35	703	0.59		56.3
Total	730.50				

SS—sum of squares; Degr. of—degrees of freedom; MS—mean squares; F—F-value.

**Table 4 plants-13-00673-t004:** *Brassica rapa* accessions with a relatively high degree of CM resistance (2021–2022).

№	Catalog Number	Name, Origin	Degree of Damage, Score			
			Pushkin *	Maykop	Murmansk	Mean
Chinese cabbage						
1	k-405	Spring Sun 60 F1, Japan	1.1 ± 0.4	1.5 ± 0.5 ^b^	1.2 ± 0.3 ^b^	1.3 ± 0.4 ^b^
2	k-406	Shantung Tropical Round, Japan	1.5 ± 0.9	0.4 ± 0.1 ^a^	1.7 ± 0.3 ^c^	1.6 ± 0.4 ^c^
3	vr.k-1558	Raioh 90 F1, Japan	0.8 ± 0.2	1.0 ± 0.6 ^b^	1.0 ± 0.0 ^b^	0.9 ± 0.4 ^a^
4	vr.k-1574	Megapolis F1, Russia	1.3 ± 0.8	1.2 ± 0.9 ^b^	1.6 ± 0.1 ^c^	1.3 ± 0.9 ^b^
Mean			1.5 ± 0.7	1.6 ± 0.8	1.6 ± 0.3	1.6 ± 0.7
Pakchoi						
5	k-316	Local, Vietnam	1.5 ± 0.9	2.2 ± 0.2 ^c^	1.0 ± 0.0 ^b^	1.7 ± 0.7 ^c^
6	k-332	Chingensay, Japan	1.5 ± 1.0	1.3 ± 0.7 ^b^	0.5 ± 0.2 ^a^	1.2 ± 0.9 ^b^
7	k-384	Local, Vietnam	1.5 ± 0.5	2.0 ± 0.1 ^c^	1.1 ± 0.2 ^b^	1.6 ± 0.5 ^c^
8	k-584	Xia Lv2 Pakchoi, China	0.5 ± 0.2	0.6 ± 0.2 ^a^	1.5 ± 0.0 ^c^	0.7 ± 0.3 ^a^
Mean			1.7 ± 0.8	1.6 ± 1.0	1.5 ± 0.7	1.6 ± 0.8
Wutacai						
9	k-695	Xiaoba Ye Ta Cai, China	1.7 ± 1.0	0.0 ± 0.0 ^a^	1.2 ± 0.3 ^b^	0.9 ± 0.6 ^a^
10	vr.k-1409	Red Tatsoi F1, USA	1.4 ± 0.5	0.0 ± 0.0 ^a^	2.4 ± 0.8 ^d^	1.0 ± 0.5 ^a^
Mean			1.8 ± 0.9	1.2 ± 1.0	1.2 ± 1.0	1.6 ± 1.0
Root turnip						
11	k-1398	Palitra, Russia	1.0 ± 0.5	2.0 ± 0.0 ^c^	1.5 ± 0.0 ^c^	1.5 ± 0.5 ^c^
Mean			1.8 ± 0.8	2.4 ± 0.7	1.6 ± 0.5	2.0 ± 0.8
Mean for the studied set			1.7 ± 0.8	1.9 ± 0.9	1.6 ± 0.6	1.7 ± 0.8
LSD_05_			-	0.22	0.10	0.12

All data are presented as mean ± SD. ^a–d^ Values with different superscripts in the column differed significantly (*p* < 0.05). *—No statistically significant differences were found. LSD_05_—Fisher’s Least Significant Difference.

**Table 5 plants-13-00673-t005:** Degree of Lepidoptera pest damage to created *Brassica rapa* lines and F1 hybrids.

№	Crop	Catalog Number of Initial Cultivar	Name, Origin of Initial Cultivar	Degree of Damage, Score		
				DBM	CM	Mean
1	Chinese cabbage	k-292	Ju Zhu, China	0.5 ± 0.2	0.5 ± 0.1	0.5 ± 0.0
2	Chinese cabbage	k-406	Shantung Tropical Round, Japan	0.3 ± 0.2	0.1 ± 0.05	0.2 ± 0.1
3	Chinese cabbage	vr.k-1558	Raioh 90 F1, Japan	0.2 ± 0.1	0	0.1 ± 0.05
4	Pakchoi	k-195	Local, China	0	0	0
5	Pakchoi	k-332	Chingensay, Japan	0	0.1 ± 0.03	0.05 ± 0.01
6	Pakchoi	k-538	Green Boy, China	0.4 ± 0.3	0	0.2 ± 0.1
7	Pakchoi	k-557	Pak Choi F1 Mei Qing Choi, Japan	0	0	0
8	Pakchoi	k-584	Xia Lv2 Pakchoi, China	0	0	0
9	Wutacai	k-271	Chiimi Jukina, Japan	0.2 ± 0.1	0	0.1 ± 0.03
10	Wutacai	k-649	Tatsoi, Japan	0.25 ± 0.07	0.25 ± 0.1	0.25 ± 0.0
11	Wutacai	k-695	Xiaoba Ye Ta Cai, China	0	0.25 ± 0.1	0.13 ± 0.05
12	Wutacai	vr.k-1409	Red Tatsoi F1, USA	0	0.5 ± 0.07	0.25 ± 0.1
13	Wutacai	vr.k-1416	Choho F1, USA	0.15 ± 0.1	1.0 ± 0.7	0.6 ± 0.4
14	Zicaitai	vr.k-1358	No name, The Netherlands	0	0	0
15	Leaf turnip	k-435	Bansei Komatsuna, Japan	0.1 ± 0.05	0.1 ± 0.01	0.1 ± 0.0
16	F1	k-271 × k-435	Chiimi Jukina × Bansei Komatsina	0.05 ± 0.01	0.25 ± 0.1	0.15 ± 0.05
17	F1	vr.k-1409 × vr.k-1416	Red Tatsoi × Choho	0	0.1 ± 0.04	0.05 ± 0.02
18	F1	vr.k-1416 × k-435	Choho × Bansei Komatsuna	0	0.25 ± 0.1	0.13 ± 0.07
19	F1	vr.k-1409 × k-271	Red Tatsoi × Chiimi Jukina	0	0	0
20	F1	vr.k-1409 × k-435	Red Tatsoi × Bansei Komatsuna	0	0	0
	Mean			0.11 ± 0.02	0.17 ± 0.04	0.14 ± 0.04

All data are presented as mean ± SD. DBM—diamondback moth; CM—cabbage moth.

## Data Availability

The data presented in this study are available in the article.

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
