# Peer review of "Eco-Geographical and Botanical Patterns of Resistance to Lepidoptera Insects in Brassica rapa L."

_plants, 2024, doi:10.3390/plants13050673_

Round 1

Reviewer 1 Report

Comments and Suggestions for Authors

Author Response

The authors thank the reviewer for the thorough evaluation of our manuscript and the valuable comments and suggestions. Please see the attachment

Reviewer 2 Report

Comments and Suggestions for Authors

The authors present a fairly extensive study in which the resistance of seven cultivars of Brassica rapa towards predation by the lepidopterans, diamondback moth (DBM) and cabbage moth (CM), in three different locations in Russia, representing three different climates, is undertaken. This manuscript possesses some merits, but needs extensive editing for English language / grammar.

Among the methodological errors that I found, the following should be addressed:

-how do the authors distinguish between damage done by cabbage moth, and by diamondback moth? Some explanation of this in the methods and materials section would be good, and also a general elaboration of the methods and materials section with inclusion of more experimental detail.

-Table 3 contains several abbreviations which should be explained, also, what do the commas in the numbers (eg. Place, 10,05 in the first row) signify? This could be explained and would make the data easier to understand to someone without experience in this field.

-there are four other items in the discussion that I feel should be addressed; I don’t know if they are methodological errors or simply grammatical errors. These are:

            -line 255, “rapeseed”; resistance of Brassica oleracea to cabbage aphid is understood, but resistance towards rapeseed? If rapeseed is a plant, how is Brassica oleracea resistant to it?

            -line 260, “Chinese cabbage Hydra” seems difficult to understand. Is Hydra a sub-variation of Chinese cabbage? An extra sentence to elaborate this would be extremely useful.

            -line 288, “A weak defeat by leaf-eating cabbage moth…” What is meant by “A weak defeat”? That cabbage moth and diamondback moth easily overcome white cabbage herbivore defenses? This sentence could be made a little more clear.

            -line 297, “heariness” is not a word, maybe “hardiness” is what the authors were looking for?

As far as English language / grammatical issues, the following should be corrected:

-line 11 and throughout the document, the word “and” should be added before the last item in a list. For instance, line 11 should read “…of the distribution of pests, and about the botanical confinement…”

-line 20, substitute “these insects” for “them”

-line 24, “…Arctic zones in 2021, while in 2022…” would be a better phrasing.

-line 56, “…and legume plants species, in total on the plants of…” would be a better phrasing.

-line 85, Bt; the complete word, Bacillus thuringiensis, should be added.

-line 97, “find” instead of “found”

-line 101, “visited” instead of “visit”

-line 103, “all” instead of “on”

-line 110, I would suggest “…and also the general genepool…” as an alternate phrasing.

-line 114, and in other similar places throughout the manuscript, the word “the” should be inserted before “Lepidoptera

-line 147, I would suggest “…Murmansk in 2021, while in 2022…” as an alternate phrasing.

-line 153 “crop” not “crops”

-Figure 2, “Murmansk” not “Myrmansk”

-line 189, “Maykop” not “Maikop”

-line 217, omit “of”

-line 242, omit “in” after “crops”

-line 245 and elsewhere, “harvest” not “harwest”

-line 247, suggest “…in recent years…” not “…in the recent years…”

-line 248, “human consumption” not “the human’s consumption”

-lines 262 and 263, what Cartea et al. found could be elaborated

-line 274, suggest “the greatest effect is on white cabbage” not “the most white cabbage”

-line 278, suggest “Many years of field research at VIR…” as an alternate wording.

-the paragraph beginning on line 284, I would recommend rewording to: “Among Lepidoptera pests the most damaging now for all brassicas in Russia is cabbage moth. Diamond back moth damages the leaves, and spoils the appearance of the plant, but cabbage moth very severely damages the leaf surface causing total harvest losses.”

-line 299 “weak” not “week”

-line 303, recommend “for the” instead of “of”

-line 311 “inbred” not “inbreeded”

-line 316 and subsequently throughout the paragraph, “was” or “were” should be used instead of “will be”.

Once these issues are addressed, which I feel is an extensive minor revision, I feel the manuscript could be published.

Comments on the Quality of English Language

English language / grammar needs extensive editing before the manuscript can be published. I have provided a list of editorial comments in my comments to the authors to this end.

Author Response

The authors thank the reviewer for the thorough evaluation of our manuscript and the valuable comments and suggestions. Please see the attachment.

Round 2

Reviewer 1 Report

Comments and Suggestions for Authors

Author Response

The authors thank the reviewer for the thorough evaluation of our manuscript and the valuable comments and suggestions. The corresponding in track changes in the re-submitted files.